# Adenosine Triphosphate: The Primordial Molecule That Controls Protein Homeostasis and Shapes the Genome–Proteome Interface

**DOI:** 10.3390/biom14040500

**Published:** 2024-04-19

**Authors:** Jianxing Song

**Affiliations:** Department of Biological Sciences, Faculty of Science, National University of Singapore, 10 Kent Ridge Crescent, Singapore 119260, Singapore; dbssjx@nus.edu.sg or songjianxing1000@gmail.com; Tel.: +65-65161013; Fax: +65-67792486

**Keywords:** ATP, nucleic acids, liquid–liquid phase separation (LLPS), membrane-less organelles (MLOs), intrinsically disordered regions (IDRs), protein folding, prebiotic evolution

## Abstract

Adenosine triphosphate (ATP) acts as the universal energy currency that drives various biological processes, while nucleic acids function to store and transmit genetic information for all living organisms. Liquid–liquid phase separation (LLPS) represents the common principle for the formation of membrane-less organelles (MLOs) composed of proteins rich in intrinsically disordered regions (IDRs) and nucleic acids. Currently, while IDRs are well recognized to facilitate LLPS through dynamic and multivalent interactions, the precise mechanisms by which ATP and nucleic acids affect LLPS still remain elusive. This review summarizes recent NMR results on the LLPS of human FUS, TDP-43, and the viral nucleocapsid (N) protein of SARS-CoV-2, as modulated by ATP and nucleic acids, revealing the following: (1) ATP binds to folded domains overlapping with nucleic-acid-binding interfaces; (2) ATP and nucleic acids interplay to biphasically modulate LLPS by competitively binding to overlapping pockets of folded domains and Arg/Lys within IDRs; (3) ATP energy-independently induces protein folding with the highest efficiency known so far. As ATP likely emerged in the prebiotic monomeric world, while LLPS represents a pivotal mechanism to concentrate and compartmentalize rare molecules for forming primordial cells, ATP appears to control protein homeostasis and shape genome–proteome interfaces throughout the evolutionary trajectory, from prebiotic origins to modern cells.

## 1. Introduction

For life to emerge, two key components are essential: biological fuel providing energy for various biological processes and molecules encoding/storing genetic information. Nature solved these two problems by selecting phosphorus as a key building element for all living organisms [1,2,3,4]. Adenosine triphosphate (ATP) emerged as the universal energy currency through the hydrolysis of its high-energy bonds, which are thermodynamically favorable but kinetically controlled (Figure 1a). On the other hand, phosphate esters provide an extremely stable backbone for RNA and DNA, which function to store and transmit genetic information [5]. ATP and nucleic acids are negatively charged molecules that share similar unit compositions. They are composed of nucleotides with phosphate groups, sugars, and nitrogenous bases. However, there are key differences: ATP has three phosphate groups and solely adenine as its base, whereas nucleic acids are the polymers of nucleotides which have a single phosphate group but five distinct bases (Figure 1b). ATP has been proposed to have emerged from unique chemistry in the water of the prebiotic and monomeric world, preceding the polymerization of RNA, DNA, and proteins, as well as the rise of genetically encoded macromolecular engines [6]. Mysteriously, in modern cells, although all known ATP-dependent proteins/enzymes require only micromolar concentrations, the cellular concentrations of ATP are very high, ranging from 2 to 12 mM depending on cell type [5].

Recently, liquid–liquid phase separation (LLPS) has been established as the common principle for forming membrane-less organelles (MLOs) and cellular compartments, including nucleoli, Cajal bodies, nuclear speckles, paraspeckles, histone locus bodies, nuclear gems, and promyelocytic leukemia (PML) bodies in the nucleus, as well as P-bodies, stress granules (SGs), and germ granules in the cytoplasm [7,8,9,10,11]. Noticeably, most, if not all, MLOs consist of the proteins abundant in intrinsically disordered regions (IDRs) and nucleic acids. Presently, the unique characteristics of IDRs are widely recognized as vital in facilitating LLPS through the establishment of dynamic and multivalent interactions. Unlike folded proteins such as lysozyme, which phase-separate only at high concentrations (>mM) [12,13,14], IDR-rich proteins undergo phase separation at considerably low concentrations (~μM) [14,15,16,17,18,19]. As such, although MLOs also contain globular client proteins, these proteins do not appear to drive the formation of MLOs.

IDRs exist as dynamic ensembles of conformers due to low energy barriers, making them highly sensitive to external changes. The multivalency of binding sites within IDR-rich proteins enables simultaneous interactions with multiple partners, facilitating simultaneous interactions with multiple copies of themselves (homotypic phase separation) or other biomolecules such as nucleic acids (heterotypic phase separation). LLPS involves the reversible separation of a homogeneous solution into coexisting dense and dilute phases, with dynamic exchange constantly occurring between two phases [3,4,5,6,7,14,15,16,17,18,19,20]. However, dynamic liquid droplets may transform into solid-like structures, leading to the formation of cytoplasmic inclusions or amyloid fibrils associated with various diseases [7,8,9,10,11,12,13,14,15,16,17,18,19,20,21,22]. LLPS has also been suggested to have been a pivotal mechanism during the primordial evolution to concentrate and compartmentalize various rare molecules, which were presumably scarce in the aqueous pools of the early Earth, thus enabling the acquisition of the complexity essential for forming primordial cells [23].

By contrast, the exact effects and high-resolution mechanisms of ATP and nucleic acids on LLPS are only just starting to be understood. Previously, both RNA [24,25,26,27] and DNA [27,28,29] have been found to exert biphasic effects on the LLPS of IDR-rich proteins/peptides, namely, induction/enhancement at low concentrations but inhibition/dissolution at high concentrations. Based on polymer chemistry principles, this heterotypic LLPS was predicted to be driven by non-specific electrostatic interactions between the negatively charged phosphate backbone of RNA/DNA and the positively charged arginine/lysine residues of proteins [25]. ATP has been found to function as a hydrotrope, preventing/dissolving the LLPS, aggregation, and amyloid fibrillation of hydrophobic proteins [30,31], and later was suggested to serve more as a cosolute [32,33]. Beyond this, ATP has also been shown to exert biphasic effects on the LLPS of FUS [27,28], implying the existence of additional effects/mechanisms by which ATP modulates LLPS.

On a physiochemical level, ATP and nucleic acids can establish extremely intricate interactions with proteins. For example, the aromatic base rings of nucleotides can engage in at least three types of interactions with proteins: (1) π–π stacking interactions with the side chains of Arg and aromatic amino acids; (2) π–cation electrostatic interactions with the side chains of Arg and Lys (Figure 1c); and (3) hydrophobic interactions with hydrophobic patches on protein. Phosphate groups of nucleotides can also exert three electrostatic effects: (1) charge–charge interactions attracting and neutralizing positively charged groups such as the side chains of Arg and Lys residues, as well as repelling negatively charged groups; (2) hydrogen bonds forming particularly through oxygen atoms in the backbone phosphate; (3) screening effects, diminishing the strength of electrostatic interactions between charged molecules and groups [34,35,36]. Intriguingly, the triphosphate group of ATP appears to possess a unique hydration structure [37]. Indeed, recently it was found that, likely by interacting with protein hydration, ATP can antagonize the crowding-induced destabilization of the human eye-lens protein γS-crystalline at a ratio of 1:1 (ATP:crystalline) [38]. Most unexpectedly, ATP and triphosphate have the capacity to energy-independently induce protein folding with the highest efficiency known so far [39]. On the other hand, almost all cellular functions of nucleic acids manifest by interaction with proteins, including their involvement in LLPS [40,41,42]. So, a question of fundamental interest arises: is ATP able to mediate the interfaces between proteins and nucleic acids?

This review therefore aims to summarize recent studies on the mechanisms of ATP and nucleic acids and their interplay in modulating the LLPS of FUS, TDP-43, and SARS-CoV-2 nucleocapsid (N) proteins by NMR, which is powerful in elucidating high-resolution mechanisms of LLPS [43]. Human FUS and TDP-43 proteins, as well as SARS-CoV-2 viral nucleocapsid (N) proteins, all contain both folded nucleic-acid-binding domains and large IDRs (Figure 1d–f). FUS and TDP-43 are RNA-binding proteins (RBPs) with a unique domain structure featuring intrinsically disordered regions (IDRs), including prion-like domains (PLDs) prone to aggregation and cellular toxicity [44,45,46]. They play crucial roles in transcription, splicing, RNA processing, DNA repair, and stress granule (SG) formation. Mutated FUS and TDP-43 can mislocalize and aggregate, contributing to neurodegenerative diseases like ALS, FTD, Alzheimer’s, Parkinson’s, and Huntington’s. Therefore, understanding their LLPS and modulation is key to unraveling these diseases and has clinical implications for new diagnostics, prognostics, and drug development. On the other hand, the SARS-CoV-2 nucleocapsid (N) protein is one of four viral structural proteins which not only packages genomic RNA (gRNA), but also suppresses the immune system and manipulates host cell machinery to enhance viral infection. It has low mutation rates, with 91% identity to SARS-CoV-1, making it a key candidate for drug development compared to the spike protein [47,48,49]. LLPS has recently been identified as a mechanism underlying its diverse functions, which are dependent on binding to viral/host-cell nucleic acids. Despite limited understanding, its interaction with gRNA is crucial for RNA genome packaging, a challenge given SARS-CoV-2’s large genome (~30 kb). Therefore, any small molecules modulating N protein–nucleic acid interactions could disrupt viral life cycle steps, potentially becoming anti-SARS-CoV-2 drugs [50,51,52,53,54,55,56]. Furthermore, accumulating evidence indicates that, via LLPS, proteins and nucleic acids form condensates/compartments in infected cells. This is also seen in other viruses, including measles [57], HIV [58], influenza [59], Hendra [60], vesicular stomatitis [61], Borna disease [62], and rabies [63]. These structures are known as viral factories, viroplasms, or viral replication centers [64]. They play crucial roles in virus–host interactions and the viral life cycle, encompassing genome replication, gene expression, and antiviral immune response [55,56,57,58,59,60,61,62,63,64]. Understanding the pivotal roles of viral LLPS not only reveals previously unknown principles of host–virus interactions and the viral life cycle, but also offers a new direction for the development of antiviral strategies and drugs, an area that remains largely unexplored.

This review primarily focuses on illustrating the biophysical principles and forces that govern how ATP and nucleic acids interplay to modulate LLPS. Briefly, ATP binds to pockets within nucleic-acid-binding interfaces of folded domains adopting different structural folds. Strikingly, both ATP and nucleic acids have been shown to mainly interact with Arg/Lys residues within IDRs. In this context, in addition to exerting extremely complex effects on proteins, ATP also interplays with nucleic acids to interact with proteins, including to modulate LLPS by competitively binding to overlapping pockets of folded nucleic-acid-binding domains and Arg/Lys residues within IDRs of human and viral proteins. Therefore, ATP appears to not only act as a primordial molecule to control protein homeostasis—including inducing folding, preventing aggregation, and mediating LLPS, essential for forming primordial cells—but has also shaped protein–nucleic acid interfaces since the dawn of life, the probiotic evolution.

## 2. ATP Binds Folded Nucleic-Acid-Binding Domains

### 2.1. ATP Binds the Nucleic-Acid-Binding Pocket of the FUS RRM Domain

FUS, a 526-residue protein, is composed of an N-terminal low-sequence complexity (LC) domain (residues 1–267), followed by an RNA-recognition motif (RRM) domain (residues 285–370) and a C-terminal LC domain enriched in RG/RGG motifs (residues 371–526) (Figure 1d). FUS is prone to aggregation, which is extensively linked to various neurodegenerative diseases, including ALS and FTD [19,21,22,44,45,46]. The FUS RRM exhibits versatile nucleic-acid-binding capacities in both sequence-specific and sequence-independent manners [65,66,67,68,69,70]. It stands out as one of the most prevalent domains in eukaryotes, commonly found in heterogeneous nuclear ribonucleo-proteins (hnRNPs) that engage in direct DNA and RNA interactions through one or several RRM domains. Structurally, the FUS RRM shares an overall fold common to other RRMs, evident in structures of its free state [65] or when bound to RNA [66,67]. This fold comprises a four-stranded β-sheet and two perpendicular α-helices. Notably, the FUS RRM possesses a unique and extra-long positively charged “KK” loop crucial for nucleic acid binding (I of Figure 2a). Intriguingly, the FUS RRM domain displays heightened backbone dynamics and relatively low thermodynamic stability with a melting temperature (Tm) of only 52 °C [69]. In particular, the FUS RRM is characteristic of irreversible unfolding and capable of spontaneous self-assembly into amyloid fibrils, which is essential for cytotoxicity [68,69,70].

In a recent investigation [70], ATP was found to bind the FUS RRM domain with a dissociation constant (Kd) of 3.8 mM (II of Figure 2a). This binding occurs at a pocket overlapping with the physiological RNA/ssDNA binding interface. Structural analysis of the ATP-RRM complex revealed that both the adenine aromatic ring and triphosphate chain engage in crucial interactions with specific residues of the FUS RRM. Specifically, ATP occupies a pocket formed by 10 residues distributed across the entire sequence (II of Figure 2a). The aromatic ring of ATP resides in a relatively hydrophobic surface pocket within the FUS RRM domain, making direct contact with the side chain of Arg328, presumably through a π–cation interaction (III of Figure 2a). Meanwhile, the triphosphate chain of ATP is nestled within a pocket featuring a positively charged surface. Notably, the oxygen atoms within the triphosphate chain form two hydrogen bonds with the side chain protons of Asn323 and Thr338, respectively (VI of Figure 2a).

Subsequent NMR titrations with AMP and triphosphate (PPP) further validated the essential roles of both the adenine and triphosphate components of ATP. AMP and PPP were observed to bind to similar pockets, but their binding affinities decreased notably, resulting in Kd values of 17.2 and 12.6 mM, respectively for AMP (V of Figure 2a) and PPP (VI of Figure 2a). On the other hand, ATP, even up to 20 mM, did not influence the thermodynamic stability of the FUS RRM domain with the same Tm of 52 °C. By contrast, under the same buffer conditions, ATP at 3 mM was sufficient to inhibit RRM fibrillization, a pivotal process associated with the acquisition of cytotoxicity linked to ALS and FTD [68]. As shown in II of Figure 2a, ATP binds the pocket which is formed by residues over the entire sequence. As such, ATP binding has been proposed to reduce the structural opening of the FUS RRM, which was shown to initiate fibrillation, thus increasing the kinetic barrier for amyloid fibrillation [69].

### 2.2. ATP Binds the Nucleic-Acid-Binding Pocket of TDP-43 RRM1 and RRM2

TAR-DNA-binding protein-43 (TDP-43) was initially identified to bind TAR DNA in HIV to inhibit its transcription [71]. TDP-43 is a 414-residue multifunctional protein within the hnRNP family. It exhibits a high affinity for single-stranded DNA and RNA, engaging with over 6000 RNA species, and holds pivotal roles in governing RNA metabolism, encompassing transcription, splicing, and mRNA stability [14,21,22,45]. TDP-43 comprises several distinct domains: the N-terminal domain (NTD) spanning residues 1–80 [72,73], two RNA recognition motif (RRM) domains spanning residues 104–262 [74,75,76], and a C-terminal PLD domain abundant in Gln/Asn/Ser/Gly over residues 265–414 (Figure 1e). Unlike the FUS RRM, TDP-43’s tandem RRM domains lack the “KK-loop” (I of Figure 2a).

Recently, it was demonstrated that ATP also binds to the pockets within the conserved nucleic-acid-binding interfaces of both RRM1 and RRM2 of TDP-43 [77,78,79], displaying notably distinct affinities (Kd of 2.6 mM for RRM1 and 13.9 mM for RRM2). Within the ATP-RRM1 complex, the triphosphate chain of ATP engages in close interactions with the side chains of K145 and K114 residues, while the ATP aromatic ring fits into a positively charged pocket formed by K136, K137, K176, and K181. Specifically, the aromatic ring of ATP establishes close contacts with the side chains of K176 and K181 via π–cation interactions. Moreover, the three phosphate oxyanions of ATP create five hydrogen bonds with the backbone atoms of S144 and K145 [77]. In contrast, within the ATP-RRM2 complex (II of Figure 2b), the aromatic ring of ATP closely interacts with the aromatic rings of F194 and F231 through π–π interactions (II of Figure 2b). Interestingly, unlike the ATP-RRM1 complex, the ATP-binding pocket on RRM2 appears to exhibit a slightly negative charge. Only one hydrogen bond forms between the β-phosphate oxyanions of ATP and the backbone atom of V195. Strikingly, the structures of TDP-43 RRM1 and RRM2 in complex with ATP closely resemble those observed in their RNA complexes, underscoring ATP’s occupation of the conserved pockets within RRM1 and RRM2 for nucleic acid binding (III of Figure 2b). Indeed, further studies have shown that the mutation of crucial nucleic-acid-binding residues disrupts ATP binding. Interestingly, the ALS-linked D169G mutation in TDP-43 RRM1, which is located on the back side of both the ATP and nucleic-acid-binding interfaces, also largely reduces ATP binding affinity to a Kd of 9.3 mM [78]. However, the ATP-binding pockets of the FUS and TDP-43 RRM domains are much smaller compared to their nucleic-acid-binding interfaces. For instance, while the “KK-loop” is crucial for the nucleic acid binding of the FUS RRM, it does not directly participate in ATP binding, thus resulting in a much lower ATP-binding affinity than for nucleic acids.

ATP exhibits varying effects on the thermal stability of the TDP-43 RRM1 and RRM2 domains and impedes amyloid fibrillation [77,78,79]. Briefly, TDP-43 RRM1 and RRM2 have Tm values of 57 and 59 °C, respectively. Interestingly, ATP at 15 mM increased the Tm value of RRM1 to 60 °C, but showed no alteration in the Tm of RRM2. On the other hand, under the same conditions, ATP at 3 mM was sufficient to inhibit the formation of amyloid fibril. As such, ATP appears to inhibit the amyloid fibrillation of TDP-43 RRM domains both thermodynamically and kinetically.

Moreover, ATP has also been characterized to bind the nucleic acid binding interface of the RRM domain of human cold-inducible RNA-binding protein (CIRBP) with a Kd of 2.9  ±  1.4 mM [80]. So, an intriguing question emerges: can ATP bind to nucleic-acid-binding domains lacking the RRM fold? Recent findings reveal that ATP indeed binds the N-terminal acidic domain (AcD) of the 623-residue SYNCRIP, exhibiting a Kd of 3.1 mM (Figure 3a). The AcD lacks any sequence and structural homology for RRMs, but functions as a cryptic RNA-binding domain adopting an all-helix fold [81]. In the complex, ATP binds to an AcD pocket formed by residues situated on helices 3 and 5 (I of Figure 3a), which constitute a significant interface for miRNA binding [82,83]. Notably, the aromatic purine ring of ATP finds placement between the side chains of Arg38 and Lys78, establishing two π–cation interactions, while the triphosphate chain directly interacts with the highly positively charged AcD surface (II of Figure 3a). Additionally, an oxyanion from the β-phosphate of ATP forms two hydrogen bonds, one with the backbone atom of Lys78 and the other with the side chain atom of Gln 82. Despite the overall negatively charged electrostatic surface of AcD, ATP can still fit into the highly positively charged cavity within AcD, which coincidentally serves as a binding site for miRNA (II of Figure 3a).

### 2.3. ATP Binds NTD and CTD of SARS-CoV-2 Nucleocapsid Protein

Severe acute respiratory syndrome coronavirus 2 (SARS-CoV-2) belongs to the coronavirus family, possessing a genomic RNA (gRNA) of approximately 30 kb, encapsulated by the nucleocapsid (N) protein within a membrane-enveloped virion. This virus has instigated the ongoing pandemic, resulting in >772 million infections and >6.99 million deaths [47]. Among its four structural proteins, the nucleocapsid (N) protein stands out as the sole protein responsible not only for packaging gRNA, but also modulating the host cell’s immune response and cellular mechanisms to facilitate viral infection and replication [48,49]. Notably, LLPS has recently been identified as a key process governing the diverse functions of the SARS-CoV-2 N protein [50,51,52,53,54,55]. Crucially, most of the N protein’s functions, including LLPS, are reliant on its interaction with a wide array of viral and host-cell nucleic acids, encompassing various single- and double-stranded RNA/DNA sequences of diverse origins in both sequence-dependent and sequence-independent manners.

The SARS-CoV-2 N protein spans 419 residues and comprises distinct domains (folded N-terminal domain (NTD) and C-terminal domain (CTD)) alongside three large intrinsically disordered regions (IDRs) enriched with Arg/Lys residues (Figure 1f). Studies have affirmed the NTD as a nucleic-acid-binding domain (RBD), crucial for engaging with diverse RNA and DNA in both specific and nonspecific manners [52,53,54,55]. Previously, CTD has been recognized for driving dimerization/oligomerization to facilitate the formation of high-order structures [49,50,51,52,53,54,55]. Very recently, CTD was also shown to be a nucleic-acid-binding domain [55,84]. Surprisingly, despite the absence of ATP in viruses [5], recent findings have revealed that ATP not only binds to NTD with a Kd of 3.3 mM [52], but also to CTD with a Kd of 1.5 mM [84].

As shown in Figure 3b, the ATP-NTD complex illustrates ATP occupying a pocket within the extensively positively charged surface (I of Figure 3b). The purine ring of ATP engages in π–cation interactions with multiple Arg residues, while the oxyanions of the β-phosphate from the triphosphate chain form three hydrogen bonds with NTD residues: two with Asn8 and one with Thr9. Recent NMR studies have revealed the structures of the NTD of the SARS-CoV-2 N protein in complex with both single-stranded RNA (ssRNA) containing the sequence UCUCUAAACG and double-stranded RNA (dsRNA) with the sequence CACUGAC [53]. Intriguingly, the comparison of these structures with the ATP-NTD complex highlights that ATP indeed occupies a pocket within the large positively charged surface used by NTD to bind various ssRNA and dsRNA (II of Figure 3b). Consequently, the NTD of the SARS-CoV-2 N protein emerges as the first viral fold capable of binding ATP with a Kd of approximately mM, in addition to the RRM and AcD folds [55].

Unexpectedly, as illustrated in III of Figure 3b, ATP can also bind two distinct binding pockets of CTD comprising 12 residues [55,84]. These pockets span the dimerization interface, consisting of residues from both monomers, notably displaying a highly positive charge (IV and V of Figure 3b). Within the complex, the purine rings of ATP closely interact with the highly positive surface formed by Lys256, Arg259, and Arg262 from one monomer, along with Lys338 from another monomer, establishing π–cation and π–π interactions between the ATP purine ring and the Arg/Lys side chains. Additionally, the triphosphate groups engage in electrostatic interactions with the predominantly positive surface, constituted by Lys342. Compared to previously identified ATP-binding pockets in other domains capable of binding ATP at mM levels, the CTD pocket features a notably high density of positive charges, characterized by a cluster of Arg/Lys residues. This unique feature potentially contributes to its high binding affinity for ATP. Interestingly, the high conservation of the 12 residues across SARS-CoV-2 variants strongly suggests that ATP binding to CTD might play an indispensable role in completing its life cycle within the host cell, thus representing a key target for the design of anti-SARS-CoV-2 drugs [55,84].

In summary, unlike the weak and non-specific effects found in folded proteins lacking nucleic-acid-binding ability, such as human γ-crystalline [38], profilin-1 [70], and others [85,86], ATP generally binds to the pockets within the nucleic-acid-binding interfaces of folded domains of diverse folds with a Kd of ~mM. Structural analysis of the ATP-binding pockets in these domains elucidates the general determinants governing their interactions: (1) Both the adenine aromatic ring and triphosphate group are essential for binding specificity and affinity. (2) π–cation/π–π interactions of the adenine aromatic ring with Arg/Lys appear to hold greater strength than the π–π interactions of the adenine aromatic ring with aromatic residues. (3) ATP exhibits a higher affinity for Arg compared to Lys, as the adenine aromatic ring of ATP is capable of forming both π–cation/π–π interactions with Arg, whereas it can only engage in a π–cation interactions with Lys. (4) The number of hydrogen bonds formed between the triphosphate chain and protein residues emerges as a critical factor influencing binding affinity. (5) ATP appears to be capable of universally binding to folded nucleic-acid-binding domains regardless of their origin, as evident from its binding to the NTD and CTD of the viral N protein of SARS-CoV-2 that lacks the ability to produce ATP itself.

Intriguingly, unlike ATP binding to kinases, which may have Kd values of ~nM [87,88], ATP only binds folded nucleic-acid-binding domains with Kd values of ~mM. Theoretically, binding affinity and specificity are two crucial factors that govern how proteins interact with other molecules. Binding affinity refers to the strength of the interaction, which signifies how tightly the two molecules bind and the degree of their complex formation. On the other hand, binding specificity describes the selectivity of the protein towards its ligand, which reflects how well the protein distinguishes its specific ligand from structurally similar molecules. Although binding affinity and specificity are often interrelated, high affinity does not always guarantee high specificity, while a protein with a lower affinity can still exhibit high specificity if its binding site is highly selective. In this context, despite its very low affinity, ATP binding to folded nucleic-acid-binding domains is generally expected to be specific for several reasons: (1) ATP-binding pockets are located within nucleic-acid-binding interfaces formed by many residues distributed over entire sequences, which can only be formed upon successful folding; (2) ATP-binding curves are all saturable; (3) the replacement of ATP with AMP or triphosphate would dramatically reduce the binding affinity; (4) the mutations of binding residues such as those of the TDP-43 RRM domain disrupt the binding events.

## 3. ATP and Nucleic Acids Interplay in Modulating LLPS

### 3.1. ATP and Nucleic Acids Interplay in Modulating the LLPS of FUS IDRs

Notably, over 70% of the FUS sequence is composed of IDRs, characterized by the absence of bulky hydrophobic residues Leu, Ile, and Val (Figure 1d). These FUS IDRs can be categorized into two distinct types: the polar/aromatic residue-rich PLD (1–165), and the RG-/RGG-rich regions spanning from 166 to 267 and the C-terminal domain (371–526), which are enriched in Gly and Arg/Lys. Within the sequence range of 1 to 165, the QGSY-rich prion-like domain (PLD) exhibits significant sequence identity with the N-terminal yeast prion domain of Sup35 [89], comprising 24 Tyr residues but no Arg/Lys. In vitro systematic dissection studies have revealed that among the FUS domains, only the FUS NTD (1–267) demonstrates autonomous phase separation at low concentrations. Conversely, the isolated RRM or C-terminal domain (CTD) lack this inherent capacity [27,69].

The FUS NTD (1–267), comprising a QGSY-rich PLD and RGG1 (Figure 3a), plays a key role in promoting self-assembly into liquid-like granules, with the potential to transition into hydrogels and solid aggregates [90,91]. By contrast, the FUS PLD was observed to undergo phase separation only at higher concentrations or in the presence of dextran (I of Figure 3a), acting as a molecular crowding agent [16,27,92]. Notably, NMR investigations revealed a fascinating discovery: the FUS PLD, which lacks autonomous phase separation, unexpectedly acquires this capability at remarkably low concentrations (~1 μM) upon linkage to RGG1 to form an NTD [27,28]. Additionally, it was observed that the isolated FUS PLD could also undergo phase separation at low concentrations upon adding the isolated RGG3 region [93]. These findings strongly support the role of π–π and/or π–cation interactions involving aromatic amino acids and Arg/Lys residues as significant contributors to driving the LLPS of FUS.

A comprehensive investigation aiming to delineate the molecular interactions governing the LLPS of FUS and its domains, alongside the impact of ATP, RNA, and specific/non-specific ssDNA sequences, was conducted [27]. The examined FUS domains encompassed PLD (1–165), NTD (1–267), and CTD (371–526). The RNA sequence used was UAGUUUGGUGAU, while the ssDNAs employed were telomeric ssDNA (TssDNA) with a sequence of (TTAGGG)_4_, forming a G-quadruplex, and (TTTTTT)_4_ (T24), lacking defined secondary or tertiary structures. Notably, the FUS PLD, incapable of phase separation at low concentrations, did not exhibit induced phase separation in the presence of ATP, RNA, or ssDNA. Corresponding NMR studies highlight the absence of substantial interactions between these molecules and the FUS PLD [27].

Conversely, as detailed previously [27,28], the FUS NTD (1–267), displaying inherent phase separation at very low concentrations, experienced a monotonic dissolution of LLPS upon the gradual addition of ATP, RNA, and two ssDNA molecules (Figure 4a). Further NMR investigations unveiled the mechanism behind this dissolution, illustrating that ATP and nucleic acids achieve this effect by binding to the same set of residues within the RGG1 region. Briefly, the LLPS of the FUS NTD predominantly hinges on π–π and/or π–cation interactions between the aromatic amino acids in the PLD and the Arg/Lys residues in RGG1. In this framework, nucleic acids with multiple base aromatic rings engage in analogous interactions with the Arg/Lys residues in RGG1, and their binding affinity is governed by binding multivalency [94]. Consequently, the base aromatic rings of nucleic acids competitively displace the aromatic rings of the PLD residues from binding to Arg/Lys residues in RGG1, leading to the dissolution of the LLPS of the FUS NTD (III of Figure 4a). Through a similar mechanism, ATP utilizes its adenine aromatic ring to establish π–π and/or π–cation interactions with the Arg/Lys residues in RGG1, albeit with notably lower affinity compared to nucleic acids. At sufficiently high concentrations, the adenine aromatic ring of ATP clusters around the Arg/Lys residues in RGG1, competitively disrupting their interactions with aromatic residues within PLD, thereby causing the dissolution of the FUS NTD’s LLPS (III′ of Figure 4a). However, in contrast to nucleic acids, the triphosphate group of ATP exhibits strong interactions with water molecules. Consequently, NTD molecules bound with ATP self-assemble into large and dynamic oligomers, involving the aromatic/hydrophobic PLD residues despite the weak ATP binding (III′ of Figure 4a). Upon introducing ssDNA molecules, which possess a higher affinity for Arg/Lys residues, they competitively displace ATP molecules clustered around these residues, resulting in the disassembly of ATP-NTD oligomers. Consequently, the NTD molecules shift to binding with ssDNA (IV of Figure 4a).

The FUS CTD (371–526) does not inherently undergo phase separation under varied conditions (I of Figure 4b). However, the introduction of ATP, RNA, TssDNA, or T24 induces phase separation, followed by subsequent dissolution. For RNA/DNA, at lower concentrations, they behave as multivalent binders, utilizing their base aromatic rings and phosphate groups to bind the side chains of Arg/Lys. This binding initiates the formation of large and dynamic RNA/DNA-CTD complexes, manifesting as liquid droplets (II of Figure 4b). Yet, at higher concentrations, excessive RNA/DNA binding disrupts these large complexes, leading to the dissolution of the droplets (III of Figure 4b). Similarly, at lower concentrations, ATP acts as a bivalent binder, using the aromatic ring of its adenine group to bind to the side chains of Arg/Lys as well as its triphosphate chain to engage with Arg/Lys. Consequently, the formation of large and dynamic ATP-CTD complexes will manifest as liquid droplets (II′ of Figure 4b). However, at high concentrations, excessive ATP binding disrupts these large complexes, leading to the dissolution of the droplets (III′ of Figure 4b).

Indeed, a comprehensive analysis of NMR data on the CTD revealed strikingly similar patterns of chemical shift differences (CSD) induced by ATP, TssDNA, and T24 [27]. Notably, a significant number of residues, including 25 Arg and 4 Lys residues, displayed pronounced shifts in their HSQC peaks. This observation strongly suggests the specific binding of ATP and ssDNA to these Arg/Lys residues. Intriguingly, the binding of ATP and ssDNA to Arg/Lys residues is not overly reliant on the specific sequence housing these Arg/Lys residues, as long as the region remains disordered. These findings provide compelling evidence that Arg/Lys residues within IDRs play a pivotal role in the induction and dissolution of phase separation by ATP and nucleic acids in the FUS CTD [27].

### 3.2. How ATP and Nucleic Acids Interplay to Modulate the LLPS of the TDP-43 PLD

TDP-43 encompasses a C-terminal PLD that is rich in Gln/Asn/Ser/Gly residues spanning residues 265–414 (Figure 1e), houses nearly all known ALS-related mutations, and is implicated in the prion-like propagation associated with ALS. Notably, in contrast to prototypical PLDs like the FUS PLD (1–165), which is abundant in polar and aromatic residues (Figure 1d), the TDP43 PLD stands out for its inclusion of 10 evolutionarily conserved methionine residues, as well as a hydrophobic region spanning residues 311–343, which adopt a nascent helical conformation capable of interacting with membranes and transforming into β-like conformations and amyloid fibrils under neutral pH conditions [95,96,97,98].

Extensive investigations have delineated the pivotal role of the evolutionarily conserved hydrophobic region in driving the LLPS of the PLD, as evidenced by the observation that the deletion of residues 311–343 abolishes its ability to undergo phase separation under various in vitro conditions [99,100]. A study at pH 6.1 proposed that the LLPS of the TDP-43 PLD primarily occurs via the intermolecular self-interaction of residues 321–340, promoting a helical propensity [95]. Moreover, the arrangements of hydrophobic ‘sticker’ residues separated by flexible linkers, electrostatic interactions, and aromatic residues adjacent to Gly or Ser have also been suggested as crucial factors [101,102]. However, other studies have indicated that at pH 6.8, the nascent helical conformation of residues 321–340 transforms into a cross-β oligomer, facilitating biologically relevant LLPS and further amyloidosis [96,97]. Evidently, the oxidation of methionine residues, enhancing helical conformation, impedes cross-β assembly and LLPS [97]. Indeed, pH was shown to be a key factor for α-to-β transformation, LLPS, and the subsequent amyloidosis of the TDP-43 PLD [96,97,98].

Although the LLPS of the FUS CTD was shown to be biphasically modulated by the binding of ATP and nucleic acids to its Arg/Lys residues (Figure 4b), the high abundance of Arg/Lys residues within the FUS CTD (25 Arg and 4 Lys) makes it extremely challenging to determine the high-resolution mechanism for ATP and RNA/DNA specifically to bind these Arg/Lys residues and modulate LLPS through NMR and site-directed mutagenesis. In contrast, the TDP-43 PLD (265–414) harbors a limited number of Arg and Lys residues—Arg268, Arg272, Arg275, Arg293, Arg361, and Lys408 (Figure 4c)—offering an excellent model for investigating potential LLPS modulation by ATP and nucleic acids, as well as whether such modulation occurs via non-specific electrostatic/salt effects of ATP and nucleic acids or through specific binding to Arg/Lys residues.

Recently, two studies comprehensively examined the impact of ATP and ssDNA on the LLPS of the TDP-43 PLD and probed residue-specific interactions using NMR [99,100]. These investigations scrutinized the effects of ATP and ssDNA at different sequences and lengths (Tar32, A32, and A6) on LLPS dynamics, assessing their interactions with both wild-type and mutated PLDs. These mutations included PLD Δ(311–343), with residues 311–343 deleted but an unaltered pI, and All-K PLD, with all five Arg residues substituted by Lys, resulting in a pI of 9.6 (Figure 4c).

Notably, ATP exhibited a biphasic modulation of the TDP-43 PLD’s LLPS, inducing phase separation at low concentrations followed by dissolution at higher concentrations [99]. By contrast, PLD Δ(311–343), which is unable to phase separate under various conditions, failed to undergo LLPS or aggregation upon the addition of ATP, even at ratios as high as 1:1500 (PLD:ATP). However, ATP induced significant NMR HSQC peak shifts in the N-terminal residues of PLD Δ(311–343), resembling patterns observed in the WT PLD. In contrast, the All-K PLD showed a minimal induction of LLPS upon ATP addition, and further ATP additions led to gradual droplet dissolution. Additionally, the ATP-triggered HSQC peak shifts in the All-K PLD resembled WT PLD patterns. Nevertheless, while the WT PLD exhibited peak saturation at a ratio of 1:100 (PLD:ATP), the All-K PLD’s shifts remained unsaturated, even at a ratio of 1:500 [99]. These findings unequivocally demonstrate the “residue-specific/selective” binding of ATP to Arg or Lys residues across different TDP-43 PLD mutants, showcasing notably higher affinity to Arg than Lys. Furthermore, unlike the FUS CTD, which relies solely on ATP for LLPS modulation due to its abundance of Arg/Lys residues, the TDP-43 PLD’s response to ATP-induced LLPS is contingent on the unique presence of the 311–343 region, despite ATP binding the same residues in both WT and Δ(311–343) PLD variants.

Hence, owing to its low binding affinity, the mechanisms underlying ATP’s role in modulating LLPS seem highly contingent on the specific context. In the case of the 156-residue FUS CTD, featuring 25 Arg and 4 Lys residues, ATP’s bivalent binding to these residues directly induces LLPS by generating sizable and dynamic complexes at low ATP concentrations, followed by dissolution at higher ATP concentrations due to excessive binding (Figure 4b). Conversely, for the 150-residue TDP-43 PLD with only five Arg residues and one Lys residue, ATP binding alone seems insufficient to drive LLPS. Thus, it necessitates the coordination of other contributing factors, particularly the oligomerization of the distinctive hydrophobic region, to induce LLPS (Figure 4c).

In a follow-up study [100], three distinct ssDNA sequences were demonstrated to biphasically modulate the LLPS of a TDP-43 WT PLD, but with varying effectiveness. A6 induced maximum LLPS at a ratio of 1:3, with complete dissolution observed at 1:5. Conversely, the stepwise addition of Tar32 led to peak LLPS at a ratio of 1:0.25, followed by dissolution at 1:1 (PLD:Tar32). A32 also triggered and dissolved LLPS in a pattern similar to Tar32. NMR analysis revealed that A6 caused shifts in a limited set of HSQC peaks, resembling the pattern induced by ATP. In contrast, Tar32 induced shifts across a broad spectrum of HSQC peaks throughout the PLD sequence, and A32 exhibited a shift pattern similar to that of Tar32. Notably, in PLD Δ(311–343), A6 caused shifts of a limited set of HSQC peaks, akin to those induced by ATP. However, even at a ratio of 1:5 (PLD:A6), A6 failed to induce LLPS. Conversely, Tar32 triggered LLPS, reaching its peak at 1:0.25, with complete dissolution observed at 1:1 (PLD:Tar32). Intriguingly, A32 displayed LLPS modulation patterns highly reminiscent of Tar32. Subsequent NMR analysis revealed that the residues with significant perturbations by both Tar32 and A32 are very similar, indicating a striking similarity between their effects.

These results suggest that the presence of the hydrophobic region 311–343 does not notably influence the binding affinity of A6, A32, or Tar32 to Arg/Lys residues. However, it plays a significant role in bolstering the force driving LLPS. As such, although ATP and A6 bind to similar residues in WT PLDs and PLD Δ(311–343), they fail to induce the LLPS of PLD Δ(311–343) due to the low binding affinity and absence of an inherent driving force from the oligomerization of the residues 311–343. In contrast, Tar32/A32, which exhibit multivalent binding with stronger binding affinity, are capable of instigating and resolving the LLPS of PLD Δ(311–343), even without the intrinsic driving force.

### 3.3. How ATP and Nucleic Acids Interplay to Modulate the LLPS of the SARS-CoV-2 N Protein

The N protein of SARS-CoV-2 features three lengthy IDRs abundant in Arg/Lys residues, but lacks Tyr and Trp, having only two Phe residues within these regions. Consequently, the N protein exhibited minimal phase separation capacity in the absence of ATP and nucleic acids. Furthermore, the C-terminal folded domain (CTD) not only facilitates dimerization, but also enhances oligomerization, potentially reinforcing LLPS but also exacerbating aggregation, akin to observations in the hydrophobic region of the TDP-43 PLD. While turbidity measurement and DIC imaging revealed the biphasic effects of ATP and the nucleic acids of both specific and non-specific sequences on the LLPS of the entire N protein, the aggregation-prone complexity hinders detailed residue-specific investigations using NMR techniques [103]. However, a recent NMR characterization of a dissected fragment, N (1–249) composed of the folded NTD and two IDRs, elucidated the modulating mechanism of LLPS by ATP and a 32-mer S2m ssDNA derived from the stem-loop II motif of SARS-CoV-2 [103].

Comprehensive NMR studies reveal that S2m drives both the formation and dissolution of LLPS in N (1–249) primarily through specific interactions with the folded NTD and Arg/Lys residues within the IDRs. A speculative model was formulated to depict this process (Figure 4d). Initially, when S2m is added to the homogeneous N (1–249) solution at lower ratios (I of Figure 4d), a dynamic and multivalent interaction occurs between N (1–249) and S2m, engaging both the NTD and IDR Arg/Lys residues. This interaction results in the formation of large and dynamic complexes, visually observed as liquid droplets (II of Figure 4d). However, as the excess S2m is introduced, multiple S2m molecules bind to a single N (1–249) molecule, causing the disruption of these large and dynamic complexes and leading to the dissolution of liquid droplets into a homogeneous solution (III of Figure 4d). The residue-specific NMR data also indicate a notable interplay between ATP and S2m in regulating the LLPS of N (1–249) by binding to overlapping sites, i.e., the folded NTD and Arg/Lys regions within the IDRs, despite exhibiting significantly different affinities. Remarkably, ATP at substantially higher concentrations than S2m was capable of disrupting the S2m-induced LLPS of N (1–249) by competitively displacing S2m from binding to the proteins (III’ of Figure 4d). Detailed NMR investigations, including ATP/S2m competition experiments, elucidated this mechanism [85]. Notably, due to S2m binding to N (1–249) with a Kd of ~μM, considerably higher than ATP (with a Kd of ~mM), ATP at 20 mM remained insufficient to completely displace S2m from binding to N (1–249) [103].

In summary, experimental investigations into the modulation of the LLPS of FUS, TDP-43 PLD, and N (1–249) composed of both folded nucleic-acid-binding domains and IDRs uncover a common mechanism by which ATP and nucleic acids interplay to modulate LLPS by competitively binding to the nucleic-acid-binding interface of folded domains, as well as arginine/lysine residues within the IDRs. Previously, ATP and RNA have also been shown to interact with the RGG-rich IDRs of human cold-inducible RNA-binding protein (CIRBP) to modulate its LLPS [80]. Furthermore, a detailed NMR study also revealed that ATP can bind to enhance or reduce the phase separation of the C-terminal IDR of an RNA-binding protein, CAPRIN1 [104].

Strikingly, by combining the semiempirical quantum mechanical (SQM) method, mean-field theory, and coarse-grained molecular dynamics (CGMD) simulations, ATP has been shown to act as a bivalent or trivalent binder to inhibit and enhance the phase separation of FUS [33]. Very recently, a comprehensive study using both experimental and simulation methods revealed the unique properties and mechanisms underlying the ATP-induced phase separation of basic intrinsically disordered proteins (bIDPs). Briefly, ATP serves as a bridging agent, cross-linking bIDP chains to create mesh-like networks that manifest as liquid-like droplets. These droplets exhibit unusual physicochemical characteristics, such as very high ATP concentrations within the droplets, rapid fusion rates, low interfacial tension, and high viscosities, thus resulting in extreme shear-thinning behavior [105].

Taken together, these results suggest the following: (1) Unlike the classic high-affinity complexes formed between some folded proteins and ATP/nucleic acid, in which various residue types contribute significantly to binding, and consequently, a minor atomic alteration in ATP/nucleic acids or proteins could dramatically alter binding affinity, ATP/nucleic acids form dynamic complexes with IDRs through bivalent/multivalent binding to Arg/Lys residues within IDRs, which lack defined conformations and are highly accessible to bulk solvent and small molecules. The formation of such complexes is mainly driven by electrostatic interactions between phosphate groups of ATP/nucleic acids and side-chain cations of Arg/Lys, as well as by π–π/π–cation interactions between base aromatic rings and Arg/Lys side chains. As a result, different bases exhibit highly similar affinities to Arg/Lys residues, while RNA and ssDNA are anticipated to bind Arg/Lys residues of IDRs using the same mechanism due to their minor chemical structural differences, suggesting that these bindings might be considered to be “residue-specific/selective”. (2) On the other hand, nucleic acids differ from ATP by having multiple covalently linked nucleotides, allowing for the establishment of multivalent but rather independent binding to Arg/Lys, resulting in length-dependent affinity. Two crucial factors governing ATP-induced LLPS are the quantity of Arg/Lys residues within IDRs and the presence of additional driving forces for LLPS. Therefore, when nucleic acids possess adequate lengths and IDRs contain multiple Arg/Lys residues, they can instigate the LLPS of these IDRs, suggesting that the magnitude of nucleic-acid-induced LLPS may exceed the level of current recognition. (3) In particular, ATP, distinguished by its unique triphosphate group, demonstrates a unique capacity for interacting with water molecules and the Arg/Lys side chain. These distinctions may impact the competition between ATP and nucleic acids in binding Arg/Lys residues, as well as their abilities to modulate LLPS. (4) Notably, although nucleic acids may attain a high binding affinity to IDRs by forming multivalent but independent bindings to multiple Arg/Lys residues, such high-affinity binding appears relatively susceptible to displacement by ATP. This is due to the fact that these dynamic complexes are highly accessible, and therefore, ATP can simultaneously displace each independent binding event via the nucleotides of nucleic acids. This phenomenon may rationalize the observation that, on the one hand, a substantial proportion of IDRs possess multiple Arg/Lys residues, rendering them prone to phase separation upon induction by nucleic acids with typical lengths. On the other hand, the occurrence of MLOs is notably limited in cells. It is therefore most likely that high cellular concentrations of ATP may function to inhibit/dissolve the majority of nucleic-acid-induced phase separations of IDRs. Consequently, proteins within MLOs consistently feature folded domains with high nucleic-acid-binding affinities, enabling the establishment of interactions with nucleic acids even in the presence of high cellular concentrations of ATP. (5) Given the highly negative charges of both ATP and nucleic acids, they possess the capability to exert strong electrostatic effects on LLPS and the aggregation of IDRs. However, these effects appear to be highly context-dependent. For instance, in the presence of excessive amounts, if ATP or nucleic acids are bound to Arg/Lys residues within IDRs such as WT TDP-43 PLD, the IDR molecules acquire site-/conformation-specific associations with multiple negative charges, resulting in repulsive electrostatic interactions disrupting LLPS and preventing irreversible aggregation. In contrast, if ATP and/or nucleic acids are unable to bind to IDRs such as the TDP-43 PLD mutant with all Arg/Lys residues mutated, their non-specific screening effect dominates, acting to induce aggregation driven by the oligomerization of the unique hydrophobic region [100].

## 4. ATP Induces Protein Folding with the Highest Efficiency

Approximately 50% of the proteome needs to fold from the unfolded state (U) to the folded state (F) to fulfill its functional role [106,107,108]. Given the marginal stability of the folded state, genetic mutations may induce destabilization, resulting in misfolding and aggregation within cellular environments—a prevalent pathological characteristic associated with aging and neurodegenerative diseases [109,110,111]. Modern cells employ supramolecular machinery, energetically fueled by ATP, as the primary mechanism to handle challenges related to protein folding and misfolding/aggregation [111]. However, the modulation of protein folding during primordial evolution, predating the emergence of complex machineries, remains an unresolved question. Notably, previous studies have demonstrated that ATP can exert proteome-wide control over the solubility and aggregation of proteins both in vitro and in vivo [112,113]. Nevertheless, the direct influence of ATP on the equilibrium of protein folding, a fundamental event in protein homeostasis, hitherto remains unknown.

Recently, it was discovered that ATP can induce the folding of ALS-causing mutants of human profilin 1 (hPFN1) (I of Figure 5a) and CuZn-superoxide dismutase 1 (hSOD1) (II of Figure 5a) [39]. C71G-hPFN1 coexists in both folded and unfolded states, with respective populations of 55.2% and 44.8%, undergoing dynamic exchange at a rate of 11.7 Hz. Nascent hSOD1, lacking metal cofactors and disulfide bridges, remains entirely unfolded, devoid of any stable secondary and tertiary structures. Through the NMR visualization of ATP and 11 related compounds, it has been demonstrated that ATP fully converts C71G-hPFN1 into the folded state at a ratio of 1:2 (C71G:ATP), while inducing nascent hSOD1 into two co-existing states at a ratio of 1:8 (hSOD1:ATP) (III of Figure 5a), the highest efficiency known so far.

Unexpectedly, the remarkable inducing capacity of ATP comes from its triphosphate moiety, previously proposed as a pivotal intermediate in prebiotic chemical evolutions, contributing to the generation of building units for constructing primitive cells and potentially influencing the origin of life [1,2,3,4]. However, it is noteworthy that triphosphate exhibits a strong ability to induce aggregation. The inducing capacity can be ranked as follows: ATP = ATPP = PPP > ADP = AMP−PNP = AMP−PCP = PP. Meanwhile, AMP, adenosine, P, and NaCl demonstrate no conversion. Surprisingly, trimethylamine N-oxide (TMAO), a well-known molecule with a general capacity to induce protein folding [108], fails to exhibit any detectable induction of the folding of C71G-hPFN1, even at a ratio of 1:2000 (C71G:TMAO), where protein aggregation occurs.

For nascent hSOD1, 12 cations (Na^+^, K^+^, Ca^2+^, Zn^2+^, Mg^2+^, Mn^2+^, Cu^2+^, Fe^2+^, Ni^2+^, Cd^2+^, Co^2+^, and Al^3+^) have been evaluated, but only Zn^2+^ and Fe^2+^ demonstrate the ability to induce folding with a coexistence of unfolded and folded states at a ratio of 1:20 (hSOD1:cation) [114]. Notably, ATP exhibits a distinct capacity to induce folding, leading to the coexistence of unfolded and folded states at a much smaller ratio of 1:8 (hSOD1:ATP) [39]. In contrast, TMAO fails to induce the folding of hSOD1, even at a ratio of 1:1000 (hSOD1:TMAO), where protein aggregation occurs. Mechanistically, a comparison of the conformations and dynamics of ATP- and Zn^2+^-induced hSOD1 folded states suggests that ATP and triphosphate induce folding by enhancing the intrinsic folding capacity encoded in the protein sequences. Specifically, ATP and triphosphate have been proposed to induce folding mainly by interacting with and displacing water molecules that hydrogen-bond with the unfolded protein [39].

Therefore, nature appears to select triphosphate as a central intermediate in prebiotic evolution, possibly also due to its notably high efficiency in inducing protein folding. Intriguingly, polyphosphates may employ a similar mechanism to triphosphate to generally facilitate protein folding. Consequently, inorganic polyphosphates persist as primordial chaperones in some single-cell organisms [115,116]. Nevertheless, owing to the propensity of high triphosphate concentrations to induce aggregation in unfolded or partially unfolded protein states, modern cells maintain extremely low concentrations of triphosphate. Nature marvelously creates ATP that possesses three integrated abilities: to effectively induce protein folding, inhibit aggregation, and enhance thermodynamic stability by joining adenosine with triphosphate [39].

## 5. ATP Action from Prebiotic Evolution to Modern Cells

A recent study suggests that ATP emerged through the phosphorylation of ADP via a universally conserved intermediate, acetyl phosphate (AcP), during the early stages of biochemical evolution [6]. Intriguingly, the highest ATP yield was observed in highly unsalted water, such as HPLC-grade water, at an optimal pH of approximately 5.5–6, while the yield significantly diminished upon the introduction of salts and alterations in pH [6]. Remarkably, the aqueous condition observed in this study [6] coincides with the solution condition in which all insoluble proteins could be solubilized [117], also akin to the conditions present in prebiotic oceans or water bodies [6,110]. Notably, AcP phosphorylated only ADP but not other nucleoside diphosphates, contributing to the universal conservation of ATP across diverse life forms.

Therefore, as illustrated in part I of Figure 5b, at the early stages of biochemical evolution, water was highly unsalted with a slightly acidic pH in which all primordial peptides/proteins were soluble [110,117]. The presence of ATP and/or triphosphate might drive the phase separation of primordial peptides/proteins with cationic residues, thus serving to concentrate these rare peptides/proteins (II of Figure 5b). Furthermore, the binding of ATP to these primordial peptides/proteins might also have protected them from aggregation upon the coming increase in salt concentration in the water [6,110]. Later, as nucleic acids emerged, primordial peptides/proteins may have become mainly bound to nucleic acids, leading to phase separation and the formation of dynamic protein–nucleic acid complexes (III of Figure 5b). Subsequently, the phase-separated droplets may have been separated from their surroundings through encapsulation within a membrane system, thereby giving rise to the formation of primordial cells (IV of Figure 5b). In primordial cells, mechanisms for the self-reproduction of proteins and nucleic acids might have evolved and established. As such, with the extensive sampling of protein sequences, some proteins become folded with the induction of ATP, while some remain intrinsically disordered. Furthermore, under modulation by ATP, protein–nucleic acid complexes also evolved into two categories: well-folded ones, as exemplified by ribosomes, and dynamic droplets for forming membrane-less organelles (V of Figure 5b).

In modern cells, genetic information is typically stored in genomes (DNA in most organisms, but RNA in some viruses). Protein–nucleic acid interactions play a central role in the expression of genetic information into proteins (proteomes). Consequently, the interfaces between genomes and proteomes are crucial for regulating, maintaining, and expressing genetic information. In eukaryotic cells, most nucleic-acid-binding proteins contain both folded and intrinsically disordered domains (Figure 6a). The binding affinity and specificity of these domains for nucleic acids vary widely, depending on the proteins and their target DNA/RNA sequences. Folded domains can have binding affinities (Kd) for nucleic acids ranging from picomolar (pM) to millimolar (mM). On the other hand, RG/RGG-rich motifs within intrinsically disordered regions (IDRs) of over 1700 human proteins were previously identified as nucleic acid binders [118,119], while recent research indicates that Arg/Lys residues within IDRs are sufficient to bind nucleic acids, with affinity highly dependent on binding valency [100].

As ATP closely resembles nucleotides, the building blocks of nucleic acids (Figure 6b), it can bind to pockets within the nucleic-acid-binding interfaces of folded domains. Consequently, ATP competes with nucleic acids for binding to these proteins. However, whether ATP can displace nucleic acids from these proteins depends heavily on the ratios of binding affinity and concentrations among ATP, nucleic acids, and proteins. Additionally, ATP seems capable of readily displacing nucleic acids from binding to Arg/Lys within IDRs because nucleic acids bind Arg/Lys to form dynamic and accessible complexes in a multivalent manner, but with each binding event occurring rather independently. Strikingly, it was demonstrated that ATP could even directly interact with and destabilize the CAG-RNA hairpin structure, thus modulating RNA–protein interactions [120]. Therefore, ATP appears to play a central role in shaping the genome–proteome interface (Figure 6c) by modulating the formation of folded protein–nucleic acid complexes and IDR nucleic acid membrane-less organelles.

ATP is composed of both adenosine and a triphosphate group with a unique hydration property [37], and is thus capable of exerting extremely complex effects on proteins and nucleic acids, including hydrotropic, cosolute, and/or screening effects at high concentrations in a context-dependent manner (Figure 6b). The triphosphate group in ATP can directly alter protein hydration even at very low molar ratios, counteracting the crowding-induced destabilization of human eye-lens γ-crystalline at 1:1 (crystalline:ATP) [38] and enhancing the protein folding of C71G-hPFN1 at 1:2 (C71G:ATP) and hSOD1 at 1:8 (hSOD1:ATP) [39]. In modern cells, ATP not only serves as the energy currency for numerous biological processes, but also energy-independently regulates protein homeostasis. The findings reviewed here suggest that ATP also sits at the genome–proteome interface, energy-independently maintaining and controlling the expression of genetic information (Figure 6c).

In the future, two areas of great interest need to be addressed, but they come with fundamental challenges. The first is that understanding the multifaceted capacities of ATP requires decoding the core mechanisms by which ATP mediates protein hydration. However, achieving this requires a significant breakthrough in experimentally and computationally deciphering the abnormality of water [121,122,123,124]. The second relates to revealing the biological consequences of ATP’s effects. This is central but also fundamentally challenging. This is due to several factors: (1) ATP’s involvement in a plethora of energy-dependent and energy-independent processes that are interconnected or coupled; (2) ATP has emerged in prebiotic chemistry, implying that its energy-independent effects might have been extensively integrated into various cellular processes during evolution; and (3), the primary challenge, the fact that biological function represents an “emergence” from complex systems/networks of biochemical/biophysical interactions that cannot be understood or predicted solely by analyzing the individual components of the systems/networks, largely due to non-linearity [125].

## Figures and Tables

**Figure 1 biomolecules-14-00500-f001:**
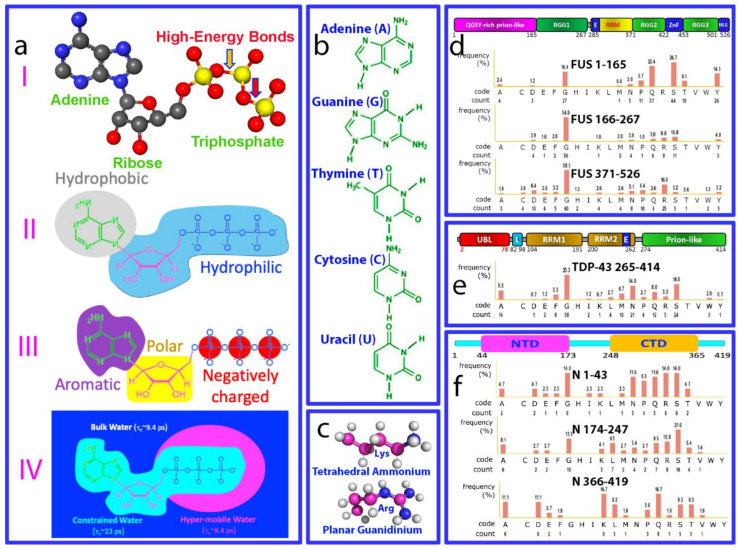
Key properties of ATP and nucleic acids for modulating LLPS of FUS, TDP-43, and SARS-CoV-2 nucleocapsid proteins. (**a**) ATP has unique structural properties and thus may act as (I) energy currency by hydrolysis of high-energy bonds; (II) a biological hydrotrope with the presence of hydrophobic adenine and hydrophilic ribose and triphosphate; (III) a bivalent binder via the aromatic purine ring and highly negatively charged triphosphate chain; and (IV) a hydration mediator resulting from its unique hydration structure previously derived from the results of microwave dielectric spectroscopy, modeled to contain “constrained water” with a dielectric relaxation time (τc) of ~23 ps, as well as “hyper-mobile water” with a τc of ~8.4 ps, even smaller than that of bulk water (9.4 ps). Chemical structures of five nitrogenous bases in DNA and RNA (**b**), and of Arg and Lys side chains (**c**). Amino acid compositions of intrinsically disordered regions (IDRs) of FUS (**d**), TDP-43 (**e**), and SARS-CoV-2 nucleocapsid (N) proteins (**f**).

**Figure 2 biomolecules-14-00500-f002:**
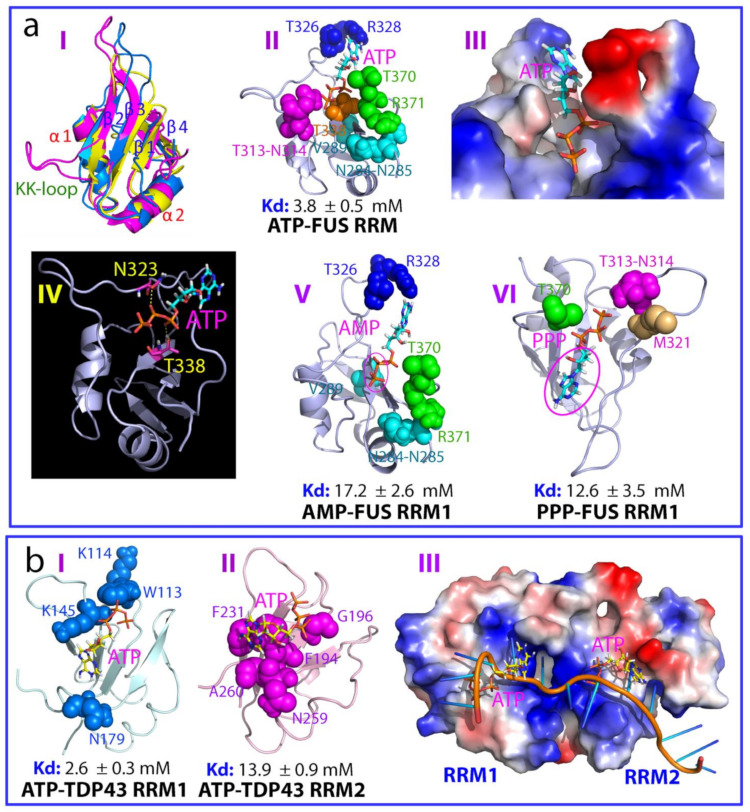
ATP binds nucleic-acid-binding pockets of FUS and TDP-43 RRM domains. (**a**) (I) Superimposition of FUS RRM (purple), TDP-43 RRM1 (blue), and RRM2 (yellow) with secondary structures labeled. (II)–(VI) ATP/AMP/PPP complexes of FUS RRM domain. (**b**) ATP complexes of TDP-43 RRM1 and RRM2 domains.

**Figure 3 biomolecules-14-00500-f003:**
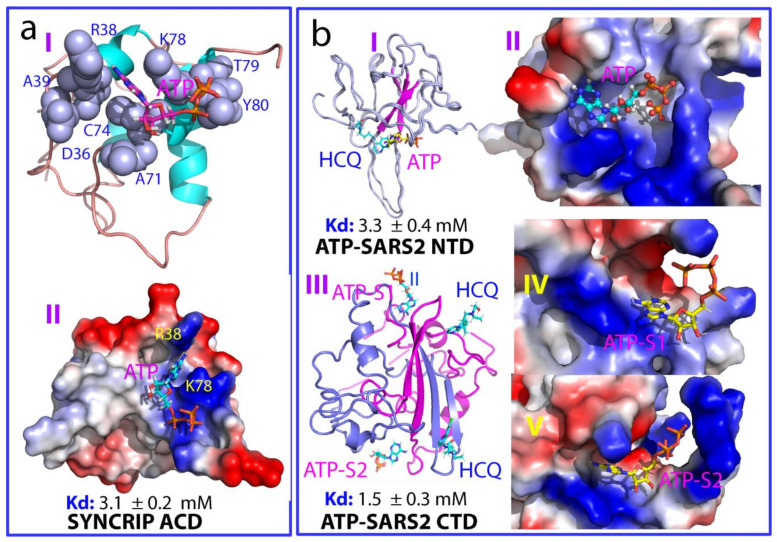
ATP binds nucleic-acid-binding pockets of folded domains beyond RRM. (**a**) ATP complex of the acidic domain (AcD) of Syncrip. (**b**) ATP complexes of N-terminal domain (NTD) and C-terminal domain (CTD) of SARS-CoV-2 nucleocapsid protein, both of which are bound with hydroxychloroquine (HCQ). CTD is a homo-dimer with two monomers colored in blue and purple, respectively.

**Figure 4 biomolecules-14-00500-f004:**
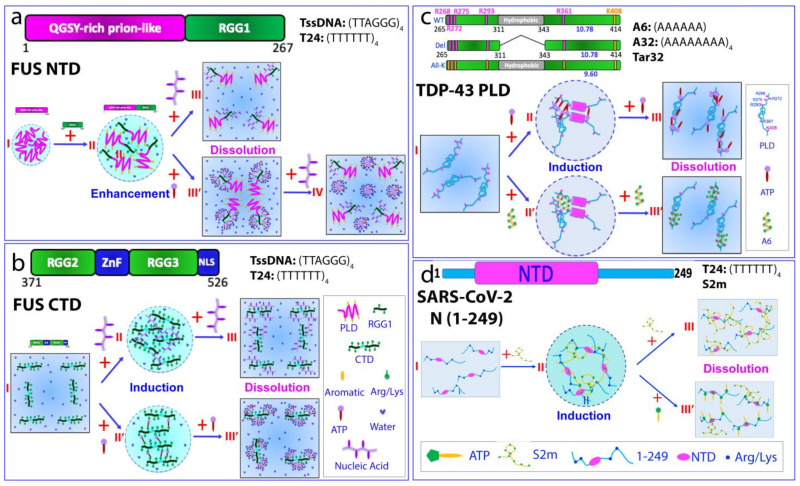
Mechanisms of interplay between ATP and nucleic acids to modulate LLPS. Speculative models to illustrate how ATP and nucleic acids interplay to modulate LLPS: (**a**) FUS NTD; (**b**) FUS CTD; (**c**) TDP-43 PLD and its mutants including the ones with residues 312–342 deleted and with all five Arg mutated to Lys; (**d**) SARS-CoV-2 N (1–249).

**Figure 5 biomolecules-14-00500-f005:**
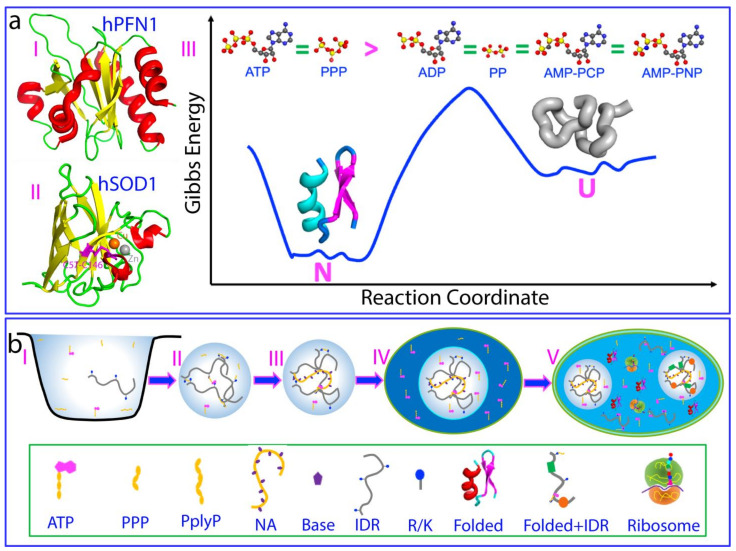
ATP enhances protein folding and is proposed to have contributed to the origin of life. (**a**) ATP energy-independently induces protein folding with the highest efficiency known so far. (**b**) ATP plays key roles in controlling protein homeostasis and interactions between nucleic acids and proteins throughout the evolutionary trajectory from prebiotic origins to modern cells.

**Figure 6 biomolecules-14-00500-f006:**
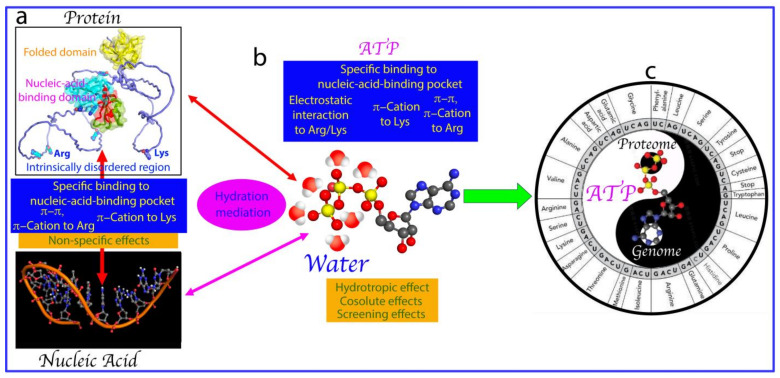
ATP shapes the genome–proteome interface. (**a**) Major interactions between nucleic acids and proteins composed of intrinsically disordered regions (IDRs) and folded domains including the nucleic-acid-binding domain. (**b**) Major effects that ATP exerts on proteins composed of intrinsically disordered regions (IDRs) and well-folded domains including the nucleic-acid-binding domain. (**c**) ATP sits in between the genome and proteome in modern cells to regulate various biological functions.

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
