# Peer review of "Adenosine Triphosphate: The Primordial Molecule That Controls Protein Homeostasis and Shapes the Genome–Proteome Interface"

_biomolecules, 2024, doi:10.3390/biom14040500_

Round 1
Reviewer 1 Report
Comments and Suggestions for Authors
This work provides synthetic view on the different mode of interaction between ATP and model IDPs that are known to drive LLPS.
In general this is a very good and valuable peace of work. I only have some minor concerns:
1. Page 3: “Furthermore, accumulating…. Immune response (55-64)”. Please, rearrange the citation. Instead of providing the all together for the whole fragment, please assign appropriate citation to the information in the text.
2. Figure 1: In my opinion section b in definitely not needed.
3. Figure 1: please remake the structures so that the color-coding for particular atoms is applied. I a and c - the colour code of the atoms do not correspond, but is should.
4. Figure 1 d, e, f – too small, and difficult to read. Also the style of the domain structure should be uniformed.
5. Figure 1 description: b , c before the information, whereas d, e, f after – please change it
6. Page 5, second line: …(RRM) domain. The “domain” is not needed.
7. VI of Figure 2a. it should be IV not VI and why the background is black.
8. Page 7: … 265-414 (Figure 1e) - residue number do not match those in the text.
9. Page 7: “Morover , the… S144 and K145 (I of Fig. 2b)….” These residues are not indicated in the image.
10. It should be ssDNA , not SsDNA, and dsRNA instead of DsRNA. Also the valence of the ions, please provide the in upper index.
Author Response
Reviewer 1
This work provides synthetic view on the different mode of interaction between ATP and model IDPs that are known to drive LLPS.
In general this is a very good and valuable peace of work. I only have some minor concerns:
- Page 3:“Furthermore, accumulating…. Immune response (55-64)”. Please, rearrange the citation. Instead of providing the all together for the whole fragment, please assign appropriate citation to the information in the text.
Response: thanks so much for the kind comment. I have revised it accordingly.
- Figure 1: In my opinion section b in definitely not needed.
Response: thanks for the kind comment.
For hardcore biochemists and structural biologists, this is basic knowledge. However, for many biologists, this subfigure appears to be essential for illustrating the minor differences in the bases of nucleic acids, which is a key foundation for the current review. Indeed, in the review reports for my previous research articles, some reviewers appeared to think DNA and RNA have very significant different structures including bases.
- Figure 1: please remake the structures so that the color-coding for particular atoms is applied. I a and c -the colour code of the atoms do not correspond, but is should.
Response: thanks for the kind comment. They have different color codes as they were generated by different softwares.
- Figure 1 d, e, f – too small, and difficult to read. Also the style of the domain structure should be uniformed.
Response: thanks for the kind comment. Fortunately, as the article will be published electronically on-line, the readers can access and view high-resolution TIF files.
- Figure 1 description:b , c before the information, whereas d, e, f after – please change it
Response: thanks for the kind comment. I have revised them accordingly.
- Page 5,second line: …(RRM) domain. The “domain” is not needed.
Response: thanks for the kind comment. I have removed it.
- VI of Figure 2a. it should be IV not VI and why the background is black.
Response: thanks for the kind comment. The black background is chosen to allow the visualization of the hydrogen boning which is represented by yellow lines.
- Page 7: … 265-414 (Figure 1e) -residue number do not match those in the text.
Response: thanks for the kind comment. The prion-like domain is over 274-414. However, in the previous experimental studied, the constructs also included the linker between RRM2 and PLD, thus being 265-414.
- Page 7: “Morover , the… S144 and K145 (I of Fig. 2b)….” These residues are not indicated in the image.
Response: thanks for the kind comment. I have revised it by referring to the detailed description in the original literature (77).
- It should be ssDNA , not SsDNA, and dsRNA instead of DsRNA. Also the valence of the ions, please provide the in upper index.
Response: thanks for the kind comment. I have corrected them all.

Reviewer 2 Report
Comments and Suggestions for Authors
I read with interest the review on the role of ATP in the formation of membraneless organelles through liquid-liquid phase separation (LLPS). The work summarizes the results of LLPS studies of only three proteins (human FUS, TDP-43 and viral nucleocapsid (N) protein of SARS-CoV-2), that allows to present a detailed analysis of the processes. At the same time, being an expert in the analyzed field, the author makes deep, important generalizations and formulates the direction of future research. The review is well written and wonderful illustrated. The review will undoubtedly attract the attention of readers.
Minor comments:
1. The abbreviation LLPS is introduced on page 2 and then again twice on page 3. These should be removed.
2. I would like to draw the author’s attention to the wording of two ideas given on page 2: “Noticeably, most, if not all, MLOs consist of the proteins abundant in intrinsically disordered regions (IDRs) and nucleic acids.” and
“Unlike the folded proteins such as lysozyme, which phase separate only at high concentrations (>mM) (12-14), IDR-rich proteins undergo phase separation at considerably low concentrations (~μM) (14-19).”
In principle, everything is correct, but MLOs contain not only IDRs, but also globular proteins, which are client proteins. On the other hand, although globular proteins can in principle separate into phases, they are not drivers of the MLOs formation. I would advise the author to change the emphasis in this fragment.
Author Response
Reviewer 2
I read with interest the review on the role of ATP in the formation of membraneless organelles through liquid-liquid phase separation (LLPS). The work summarizes the results of LLPS studies of only three proteins (human FUS, TDP-43 and viral nucleocapsid (N) protein of SARS-CoV-2), that allows to present a detailed analysis of the processes. At the same time, being an expert in the analyzed field, the author makes deep, important generalizations and formulates the direction of future research. The review is well written and wonderful illustrated. The review will undoubtedly attract the attention of readers.
Minor comments:
- The abbreviation LLPS is introduced on page 2 and then again twice on page 3. These should be removed.
Response: thanks for the kind comment. I have corrected them all.
- I would like to draw the author’s attention to the wording of two ideas given on page 2: “Noticeably, most, if not all, MLOs consist of the proteins abundant in intrinsically disordered regions (IDRs) and nucleic acids.” And “Unlike the folded proteins such as lysozyme, which phase separate only at high concentrations (>mM) (12-14), IDR-rich proteins undergo phase separation at considerably low concentrations (~μM) (14-19).” In principle, everything is correct, but MLOs contain not only IDRs, but also globular proteins, which are client proteins. On the other hand, although globular proteins can in principle separate into phases, they are not drivers of the MLOs formation. I would advise the author to change the emphasis in this fragment.
Response: thanks for the kind comment. As inspired by the comment, I have added one sentence to clarify it as: “As such, although MLOs also contain globular client proteins, these proteins do not appear to drive the formation of MLOs.”

Reviewer 3 Report
Comments and Suggestions for Authors
The article by Jianxing Song is a rather ambitious review that covers ATP's role in providing energy to drive and support many processes within living cells, ATP's role facilitating the folding of a few proteins, and ATP + nucleic acids ability to modulate protein-mediated liquid-liquid phase separation (LLPS) in forming membraneless organelles. It then culminates to a section on the possibility of ATP having a key role as a primordial molecule. The sections covering the observed effects of ATP and nucleic acids on LLPS of FUS, TD-43, and N protein were well done with a high level of detail contrasting and comparing the 3 different protein systems. The other sections were brief and seemed somewhat out of place (i.e., the contrast of high detail versus light detail information). Maybe strengthen the other sections or delete them?
Other concerns:
Figure 4 had some unreadable components and is confusing / not clear.
Is the author arguing that prebiotic oceans were HPLC-grade water (pg 17) and only later there was a "coming increase of salt concentration" in the oceans (pg 18)? That seems highly unlikely, at least to this reader.
What exactly does the author mean by the "genome-proteome interface"? The genome is the genetic information in a cell, while the proteome is the protein content. What is meant by the interface? Could the author clarify this idea/concept? Is it simply that proteins interact with components of the genome, and it's those point of contact that comprise the "interface"? This could be explained better.
Author Response
Reviewer 3
The article by Jianxing Song is a rather ambitious review that covers ATP's role in providing energy to drive and support many processes within living cells, ATP's role facilitating the folding of a few proteins, and ATP + nucleic acids ability to modulate protein-mediated liquid-liquid phase separation (LLPS) in forming membraneless organelles. It then culminates to a section on the possibility of ATP having a key role as a primordial molecule. The sections covering the observed effects of ATP and nucleic acids on LLPS of FUS, TD-43, and N protein were well done with a high level of detail contrasting and comparing the 3 different protein systems. The other sections were brief and seemed somewhat out of place (i.e., the contrast of high detail versus light detail information). Maybe strengthen the other sections or delete them?
Other concerns:
Figure 4 had some unreadable components and is confusing / not clear.
Response: thanks for the kind comment. As this is a review to avoid the high similarity, detailed descriptions were not included. The readers could read the original articles if they are interested in these details.
Is the author arguing that prebiotic oceans were HPLC-grade water (pg 17) and only later there was a "coming increase of salt concentration" in the oceans (pg 18)? That seems highly unlikely, at least to this reader.
Response: thanks for the kind comment. This assumption is in fact the foundation of the discussion in the evolutionary context of Ref. 6, in which evidence was presented and discussed. Furthermore, in my previous Review (Ref. 110), I also discussed this and included key references. Therefore, I have revised the sentence as: “Remarkably, the aqueous condition observed in this study (6) coincides with the solution condition in which all insoluble proteins could be solubilized (117), also akin to the conditions present in prebiotic oceans or water bodies (6,110).”.
What exactly does the author mean by the "genome-proteome interface"? The genome is the genetic information in a cell, while the proteome is the protein content. What is meant by the interface? Could the author clarify this idea/concept? Is it simply that proteins interact with components of the genome, and it's those point of contact that comprise the "interface"? This could be explained better.
Response: thanks for the kind comment. As inspired by the comment, I have added sentences to discuss this topic as: “In modern cells, genetic information is typically stored in genomes (DNA in most organisms, but RNA in some viruses). Protein-nucleic-acid interactions play a central role in the expression of the genetic information into proteins (proteomes). Consequently, the interfaces between the genome and the proteome are crucial for regulating, maintaining, and expressing genetic information.”
